# Multifunctional aggregation network of cell nuclei segmentation aiming histopathological diagnosis assistance: A new MA-Net construction

Qiumei Pu [1]*, Jinglong Tian[1], Donghao Wei[1], Qingming Shu[2], Minghui Sun[3], Lina Zhao [3]*

1 School of Information Engineering, Minzu University of China, Beijing, China, 2 Department of Pathology, Third Medical Center of Chinese PLA General Hospital, Beijing, China, 3 Key Laboratory for Biomedical Effects of Nanomaterials and Nanosafety, Institute of High Energy Physics, Chinese Academy of Sciences, Beijing, China

* linazhao@ihep.ac.cn (LZ); puqiumei@muc.edu.cn (QP)

**Data Availability Statement:** The data used in our work can be accessed through the following link: https://doi.org/10.6084/m9.figshare.26073790.v2.

**Funding:** National Key Research and Development Program of China (2021YFA1200904 and

## Abstract

Automated diagnostic systems can enhance the accuracy and efficiency of pathological diagnoses, nuclear segmentation plays a crucial role in computer-aided diagnosis systems for histopathology. However, achieving accurate nuclear segmentation is challenging due to the complex background tissue structures and significant variations in cell morphology and size in pathological images. In this study, we have proposed a U-Net based deep learning model, called MA-Net(Multifunctional Aggregation Network), to accurately segmenting nuclei from H&E stained images. In contrast to previous studies that focused on improving a single module of the network, we applied feature fusion modules, attention gate units, and atrous spatial pyramid pooling to the encoder and decoder, skip connections, and bottleneck of U-Net, respectively, to enhance the network's performance in nuclear segmentation. The dice coefficient loss was used during model training to enhance the network's ability to segment small objects. We applied the proposed MA-Net to multiple public datasets, and comprehensive results showed that this method outperforms the original U-Net method and other state-of-the-art methods in nuclei segmentation tasks. The source code of our work can be found in https://github.com/LinaZhaoAIGroup/MA-Net.

## Introduction

According to the report of cancer data, released by the International Agency for Research on Cancer (IARC) in January 2022, there were about 19.29 million new cancer cases increased in the world in 2021 [1]. When cancer is in its early stage, early treatment can greatly improve the survival rate of patients, while reducing the incidence rate of patients and the cost of cancer treatment [2]. Histopathological diagnosis of tissues is the most reliable means of cancer diagnosis and serves as the gold standard for cancer determination in modern medical practices

2020YFA0710700), the National Natural Science Foundation of China (31971311, 12375326), the Innovation Program for IHEP (E35457U210).

**Competing interests:** The authors declare no competing interests.

[3]. Pathologists can determine the proportion of cancer cells and grade cancer by observing rich information such as cell location, density, and morphological features in digitized histo-pathological images from patient biopsies. This plays a crucial role in the diagnosis and treatment of cancer. However, analyzing digital pathology images demands extensive experience from pathologists, leading to subjectivity in manual assessments. Moreover, this task is labor-intensive, and when confronted with a large volume of digital pathology images, doctors might experience fatigue, leading to the possibility of misdiagnosis or overlooking crucial details, potentially significantly impacting subsequent patient treatment. With the assistance of computer-aided diagnostic systems, diagnostic efficiency can be significantly enhanced, yielding more precise and objective analytical outcomes. This aids pathologists in promptly devising follow-up treatment plans for patients. Image segmentation techniques can extract crucial foreground information from images while eliminating irrelevant background interference. Precise nuclei segmentation serves as the foundation for numerous subsequent pathological diagnostic tasks and is a vital component of pathology-assisted diagnostic systems.

Currently, nuclear segmentation methods can broadly be classified into two categories: traditional segmentation methods and deep learning-based segmentation methods. Traditional segmentation methods include threshold-based segmentation [4], watershed segmentation [5], morphological processing [6], etc. The threshold-based segmentation method is the simplest image segmentation technique, dividing image pixels into different categories based on predefined threshold parameters. This method disregards spatial information of pixels, making it susceptible to noise interference and resulting in less precise segmentation outcomes. The watershed segmentation method uses the difference of gray distribution and background color in the nuclear image to segment the nuclei. The morphological segmentation method is also vulnerable to the influence of background factors, resulting in fuzzy segmentation results. For example, Prakarsa et al. [7] proposed a medical blood cell image segmentation model based on the global adaptive threshold segmentation method, but some features are still lost. Gu et al. [8] proposed a marker-controlled watershed algorithm for nuclear clustering segmentation, and the segmentation effect is not obvious. Therefore, the traditional segmentation methods are still limited to segmenting the nuclei through prior knowledge, and it is difficult to obtain accurate results when segmenting the nuclear image with complex background.

Neural networks represent a statistically-driven methodology, adapting network weights through extensive data training. During the inference phase, these networks automatically extract abstract features from input images, enabling the completion of subsequent tasks like classification and segmentation. In comparison to conventional approaches, neural networks often produce segmentation outcomes characterized by higher resolution and accuracy. The Convolutional Neural Network, as known as CNN, has been widely used to deal with visual image problems, such as object detection [9,10], image segmentation [11,12], medical image processing [13,14], and so on. One of the key advantages of CNNs is their ability to generate feature maps with distinct receptive fields at different depths within the network. This enables the integration of these feature maps to gather both local and global information from images. The advantage of CNNs has sparked researchers' interest in exploring how CNN-based methods can better achieve image segmentation, so far, numerous outstanding segmentation models have been proposed, such as FCN, SegNet, Deeplab, and the U-Net series. These models have been widely applied across various fields for image segmentation tasks.

While image segmentation based on deep learning has exhibited significant advancements over traditional methods, empirical evidence demonstrates that the segmentation performance is not satisfactory for images containing minuscule cell nuclei or exhibiting low contrast between nuclei and the background. The primary reasons behind these issues can be attributed to two factors: Firstly, the inadequate feature extraction capability of the network's encoder

section makes it challenging to effectively extract pertinent information from intricate cell staining images for distinguishing between cell nuclei and background tissues. Secondly, the insufficiency of spatial information in the images obtained by the decoder is due to the down-sampling process. While downsampling can abstract more semantic features, it simultaneously leads to the loss of spatial information. Consequently, the decoder is unable to access sufficient information to generate accurate segmentation outcomes.

To address these challenges, we proposed a multifunctional aggregation network named MA-Net. The contributions of this paper are as follows:

1. In the encoder section of our model, we have integrated the Down-sampling Fusion module and the Context Extractor module to enhance the network's feature extraction capability and address the issue of missing small cell structures.

2. During the decoding phase, we leverage Attention Gates and the Up-sampling Fusion module to process feature maps, furnishing the decoder with higher-quality features. This approach results in the generation of improved segmentation outcomes.

3. By employing a composite loss function based on Binary Cross Entropy Loss and Dice Loss during training, we further enhance the segmentation efficacy for small targets. The model's performance is evaluated on two datasets, MoNuseg and TNBC. Experimental results substantiate that our proposed MA-Net outperforms other improved models in terms of cell nuclei segmentation tasks.

## Related work

The U-Net network proposed by Ronneberger et al. [15] has gained widespread adoption in various medical image segmentation tasks due to its outstanding performance. It adopts the encoder-decoder structure and skillfully combines the deep-level and shallow-level information by skipped connection (concatenation). The deep abstract information of the decoding layer makes better use of the shallow information transmitted by the encoding layer, making the image segmentation effect better. Lu *et al.* [16] applied a U-Net-based network to segment the retinal layer of optical coherence tomography (OCT) images. Sarhan *et al.* [17] used improved U-Net for the optic disc segmentation in high-precision retinal fundus images. The U-Net is also applied to other aspects of segmentation, such as cartilage and meniscus from knee MRI data [18] and lung lesions from COVID-19 chest CT scan images [19].

Ozan *et al.* [20] proposed a novel U-Net-based model consisting of the attention gate (AG), which can automatically learn to focus on target structures of different shapes and sizes. The key of AG is that it can make the model suppress irrelevant regions of input images and highlight salient features useful for a specific task. Also, AG could be easily applied to U-Net or other CNN architectures with minimal computation and high prediction accuracy. Zhou *et al.* [21] proposed a novel and more powerful UNet++ architecture. They integrated U-Nets at the depth of 1 to 4 into one architecture. UNet++ could be pruned by a deep supervision technique. The advantage of pruning is that it could reduce the inference time but decrease a little accuracy performance. Gu *et al.* [22] present a context encoder network (CE-Net). Because of its two main parts, dense atrous convolution(DAC) block and residual multi-kernel pooling (RMP) block, CE-Net enables to capture more deep-level information and retains more spatial information for medical 2D image segmentation. Xiang *et al.* [23] present a Bi-directional O-shape network (BiO-Net). The model has a simple structure that recurrently reuses the building blocks by adding no more extra blocks and parameters.

## Methodology

Our proposed MA-Net employs U-Net as the backbone, as shown in Fig 1, the network structure can be divided into four parts: the downsampling path, the bottleneck block(the bottom of the U-shaped structure), the upsampling path, and the skip connections. Previous work has typically focused on improving individual parts to enhance performance. However, we believe that improving a single part is insufficient, especially for complex scenes like nucleus segmentation. Our improvement strategy involves simultaneously enhancing all four parts. In the upsampling and downsampling paths, we optimize the feature fusion strategy to better integrate features with different receptive fields. In the bottleneck block, we further process the encoded high-level semantic features. In the skip connection structure, we use spatial attention mechanisms to better handle image details.

During the encoding process, we incorporate the Down-sampling Fusion module to integrate image spatial information (pink parts in Fig 1). The module directly performs multiple downsampling operations on the input image (via 2-stride convolution instead of max pooling) to form a structure similar to an image pyramid, thereby obtaining information at different scales of the original image. These feature maps are then connected to the U-Net encoder as additional inputs. It is mathematically shown in Eq 1.

$$DF_i^{\left(\frac{1}{2}H \times \frac{1}{2}W \times 2C\right)} = Conv_{s2}\left(DF_{i-1}^{(H \times W \times C)}\right) \ \ i = 1, 2, 3, 4 \tag{1}$$

Where function $Conv_{s2}(\bullet)$ is a stride 2 convolution operation, $DF_i^{(H \times W \times C)}$ denote the feature map at the i-th layer of the U-Net encoder.

At the end of the downsampling, we utilized the Context Extractor Module composed of dense atrous convolution (DAC) block and residual multi-kernel pooling (RMP) block to extract context semantic information and generate more deep-level feature maps. DAC is composed of dense atrous (dilated) convolutions, which can achieve a larger receptive field with fewer parameters, overcoming the limitations of pooling layers that cause loss of image semantic information [24]. The DAC consists of four parallel atrous convolution branches with different dilated rates, allowing it to extract features from different sizes and scales. RMP uses four residual convolution kernels of different sizes to encode global context information. The kernels sizes are set to 2×2, 3×3, 4×4, and 6×6. It could detect objects of different sizes on account of these different sizes of kernels.

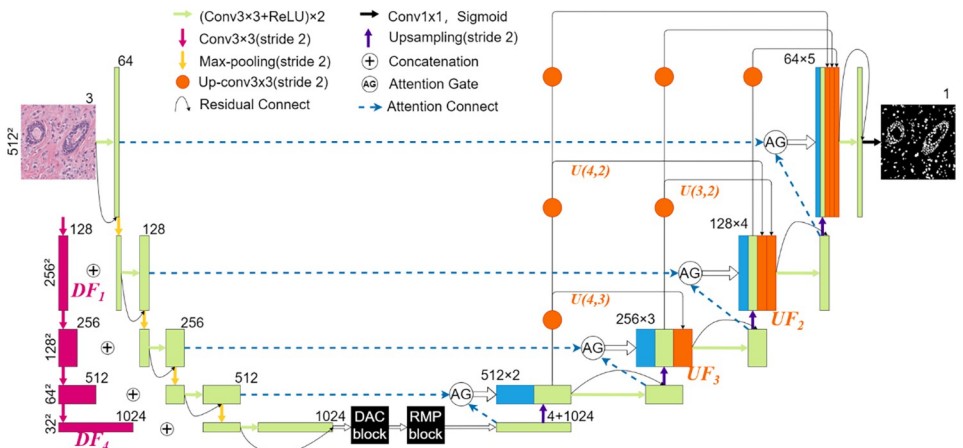

**Fig 1. Overview of the proposed MA-Net architecture.**

In the decoder part, in order to strengthen feature propagation and feature reuse, and alleviate the vanishing gradient problem, we added the Up-sampling Fusion Module. This module was inspired by Gao et al. [25] who proposed Dense-nets. The output of each upsampling is spliced with the feature map of subsequent upsampling. We employ transpose convolution with a stride of 2 in each dense connection path to ensure that the generated feature maps have appropriate dimensions and channel numbers. The calculation formula is given in Eq 2.

$$DF_i = Concat(U_{(4,i)}, U_{(3,i)}, \cdots, U_{(i+1,i)}) \ \ i = 1, 2, 3 \tag{2}$$

where function $U_{(i,j)}$ is upsampling operation to upsample featuremap from layer j to layer i, function $Concat(\bullet)$ is concatenate operation, $UF_i$ denotes an output feature map of up-sampling fusion module.

Attention Gate Module (AG) is added to the skip-connection. AG introduces spatial attention to the feature maps by assigning pixel-level weights, enhancing the response of important regions while suppressing irrelevant signals. In this way, the network can focus more on the foreground of the cell nuclei, mitigating the adverse effects of complex irrelevant background information on the segmentation results to some extent.

At the last, We replaced the convolution blocks of the original U-Net with Residual Modules. It helps alleviate the issue of information loss in traditional convolutional networks to some extent and avoids the gradient disappearance and gradient explosion problems that may be caused by the deep convolution neural network, so as to enhance the robustness of the model.

Fig 2 illustrates the structural details of the aforementioned modules. In summary, the goal of MA-Net is to retain as much image detail information as possible while improving the quality of feature maps. This aims to enhance the network's ability to recognize cell nuclei of different sizes and colors, thereby avoiding phenomena such as holes in segmented nuclei, nuclei adhesion, and the omission of small nuclei in the segmentation results.

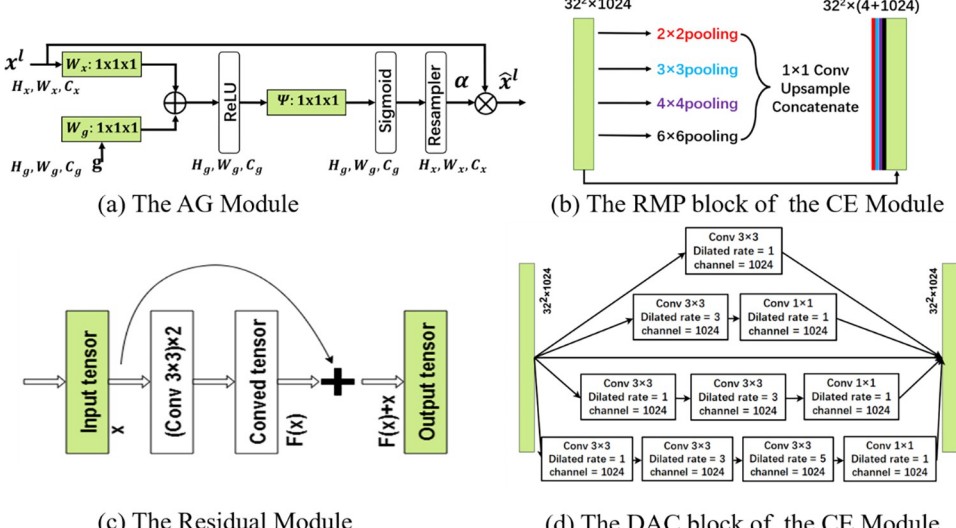

**Fig 2. Illustration details of the constructed MA-Net's main modules.**

## Experiments and results

### 4.1 Dataset

In our experiments, we utilized two publicly available cell nucleus segmentation datasets, namely MoNuSeg [26] and TNBC [27]. The Indian Institute of Technology Guwahati shared their diligently prepared MoNuSeg dataset. It includes 44 H&E stained images sampled from 7 different organs, officially using 30 images for the training set and the remaining 14 images for the testing set. We divided each image of original size $1000^2$ into 4 parts of the same size $500^2$ then resize them into $512^2$ as our method inputs, enlarging the dataset by 4 times.

The segmentation and stitching of images are commonly used data augmentation methods, such as CutMix [28] and CutOut [29]. The data augmentation method we employ is rational and effective, it serves the purpose of increasing the amount of data. Although the process of segmentation may result in some complete cells being divided into two parts, it does not have a detrimental impact on the segmentation results.

This is because both the MoNuSeg and TNBC datasets are obtained by splitting huge-sized whole-slide images (WSIs). Due to the excessively large size of WSIs, it is necessary to divide them into smaller-sized images before they can be directly input into the network, the division of cell nuclei into two halves is inevitably present in the original data of these two datasets. The image augmentation method we utilized only changes the size of the input image without altering the scaling level, thus it does not lead to any loss of information contained in the image. Consequently, this method does not have adverse effects on the segmentation results.

The TNBC dataset is comprised of 50 histopathology images at the size of $512^2$, divided into 11 groups. There are no official training sets or testing sets. So we chose 1 image from each group to make up our testing set, and the remaining 39 images for the training set. Both datasets include pixel-level annotated mask images for the cell nuclei semantic segmentation task and we set 30% of the training set as our experimental validation set.

### 4.2 Implementation details

Firstly, we conducted an ablation experiment on the MoNuSeg dataset to compare the impact of each module of MA-Net on the final segmentation results. Then we used MA-Net and other advanced methods to carry out comparative experiments respectively and obtained the experimental results.

Adam [30] optimizer would be the first choice for training most deep learning models. We used it and set its initial learning rate as 0.001 and decay rate as 0.1. The training set was not augmented, we set the batch size as 4 in both the training and the testing phases. The experimental models were trained by 100 epochs in all experiments without any pre-trained parameters and the convolution channels of the first convolution layer were set 64 to all models. The experiment was run on a single NVIDIA TESLA V100 GPU with Keras [31] and Tensorflow [32].

The loss function is of great significance to the convergence speed and final effect of the model. All models used in this experiment are end-to-end deep learning models, the segmentation results consist of two classes: foreground (cell nuclei) and background (non-cell nuclei). Essentially, it involves performing pixel-level binary classification on the image, Binary Cross Entropy(BCE) is the most common loss function for binary classification problem as shown in Eq 3. The BCE loss only focuses on the overall pixel classification accuracy and is independent of the foreground's area ratio. However, the dataset contains a significant variation in cell nucleus sizes, and the BCE loss may not be sensitive enough to accurately classify small-size nuclei. We decided to combine Dice coefficient loss [33] in Eq 4 and BCE loss as loss function

in Eq 5 to alleviate the problem of small-size targets detecting.

$$L_{BCE} = -\frac{1}{n}\sum_i (g_i \ln p_i + (1 - g_i)\ln(1 - p_i)) \tag{3}$$

$$L_{Dice} = 1 - \frac{2\sum_i p_i g_i}{\sum_i p_i + \sum_i g_i} \tag{4}$$

$$L_{BCE-Dice} = \beta L_{Dice} + (1 - \beta)L_{BCE} \tag{5}$$

In Eqs 3 to 5, $p_i$ stands for pixel-level predicted probability. $g_i$ stands for pixel-level ground truth. $\beta$ stands for the weight coefficient and we set it to 0.75 in experiments.

To evaluate the performance, we adopt a total of four evaluation indicators, which are Precision(Pre), Accuracy(Acc), Dice coefficient(Dice), and MIoU. They have been commonly and widely used to evaluate the effect of semantic segmentation.

$$Pre = \frac{TP}{TP + FP} \tag{6}$$

$$Acc = \frac{TP + TN}{TP + FP + TN + FN} \tag{7}$$

$$Dice = \frac{2TP}{2TP + FP + FN} \tag{8}$$

$$MIoU = \frac{1}{2}\left(\frac{TP}{FN + FP + TP} + \frac{TN}{FN + FP + TN}\right) \tag{9}$$

In Eqs 6 to 9, $TP$, $TN$, $FP$, $FN$ represented the number of true positive, true negative, false positive, false negative pixels, respectively.

Fig 3 shows the training and inference scheme for nuclei segmentation using our constructed MA-Net. In the training stage, we cut the training images and their corresponding ground truth images into several patches and resize them to size $512^2$, then train the MA-Net model. In the inference stage, we used the same way above to get the testing patches, then fed them to the trained MA-Net and got the segmented result.

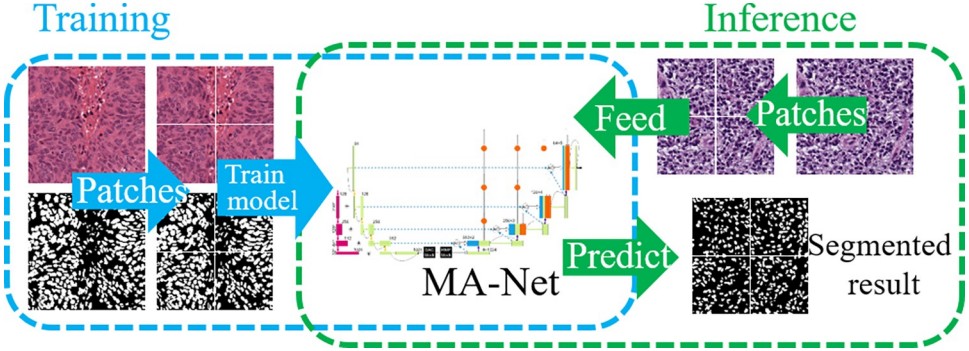

**Fig 3. Proposed training and inference to segment nuclei using MA-Net.**

## 4.3 Results and discussion

Our constructed MA-Net is composed of five extra modules based on U-Net. We conducted a simple Ablation Experiment on the MoNuSeg data set to verify the impact of these modules on the final segmentation results.

The Residual module is added to each convolution block, and its computational cost is relatively small compared with the other 4 modules, we call it the small module for the moment. The other four modules, the Attention Gate module, the Context Extractor module, the Down-sampling Fusion Module, and the Up-sampling Fusion Module, are added in fixed places with a relatively large cost of computation, which can be called big modules for the moment.

As Table 1 shows, after adding residual connections to the convolutional blocks of the original U-Net, the segmentation performance of the model has improved, on this basis, we sequentially incorporate other improvement modules into the network and evaluate the network performance. The experimental results demonstrate that the performance of the model improves regardless of which major module is individually incorporated, The improvement brought by the Attention Gate module is particularly evident. This phenomenon indicates the necessity of further processing the feature maps extracted by the encoder. By introducing spatial attention in the skip-connection structure, the response of relevant regions in the feature maps is enhanced, thereby restoring more image details during the decoding stage. Furthermore, by incorporating the upsampling fusion module and downsampling fusion module alongside the attention module, the segmentation performance is further improved, This aligns with our improvement strategy, which suggests that the performance of the original U-Net is hindered by the loss of information during downsampling. Introducing more image detail information during the decoding stage can enhance the segmentation results.

At the same time, we noticed that the DAC module does not exhibit significant improvements when used alone (combined with U-Net and residual modules as the baseline network). However, when used in conjunction with other modules, it can greatly enhance the performance of the network. The DAC module allows for the extraction of higher-level abstract features. However, the DAC module is located at the bottom of the U-shaped structure, specifically at the end of the encoder. The input feature maps of this module contain advanced semantic information but have lost a significant amount of spatial information. Using the DAC module alone cannot effectively address the issue of information loss and therefore cannot fully leverage its advantages. However, when used in conjunction with other modules designed to enhance detail information, it achieves significant segmentation results and receives the highest evaluation scores.

**Table 1. Ablation experimental results.** Res for the Residual module, AG for the Attention Gate module, CE for the Context Extractor module, Down for the Down-sampling Fusion Module, and Up for the Up-sampling Fusion Module.

| Model | Pre($\uparrow$) | Acc($\uparrow$) | Dice($\uparrow$) | MIoU($\uparrow$) |
|---|---|---|---|---|
| 1.U-Net | 0.7013 | 0.8864 | 0.7393 | 0.7235 |
| 2.U-Net+Res | 0.7162 | 0.8915 | 0.7552 | 0.7467 |
| 3.U-Net+Res+AG | 0.7332 | 0.9056 | 0.7680 | 0.7602 |
| 4.U-Net+Res+CE | 0.7185 | 0.8976 | 0.7647 | 0.7439 |
| 5.U-Net+Res+Down+Up | 0.7218 | 0.9040 | 0.7664 | 0.7523 |
| 6.U-Net+Res+AG+CE | 0.7355 | 0.9085 | 0.7734 | 0.7635 |
| 7.U-Net+Res+AG+Down+Up | 0.7416 | 0.9086 | 0.7707 | 0.7637 |
| 8.U-Net+Res+CE+Down+Up | 0.7303 | 0.9019 | 0.7692 | 0.7606 |
| 9.U-Net+ALL | **0.7803** | **0.9204** | **0.7936** | **0.7826** |

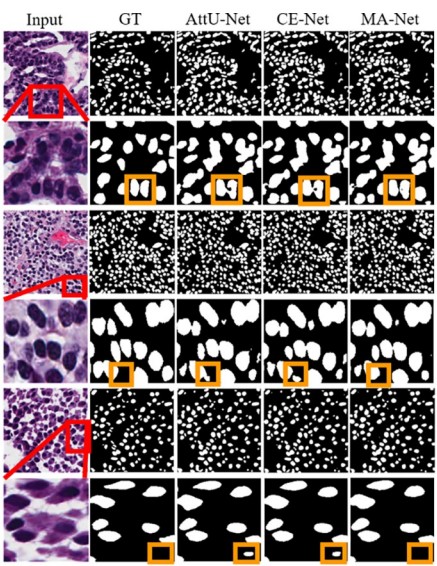 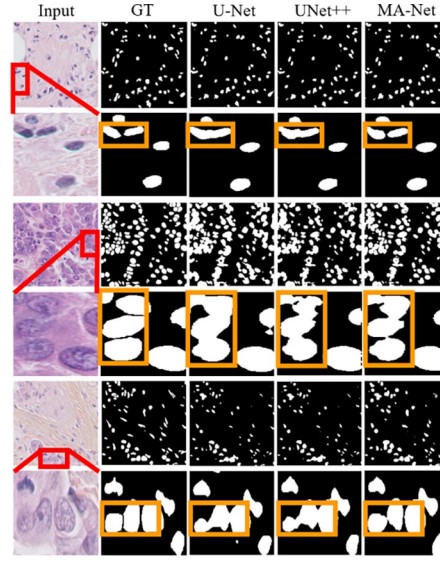

(a) Comparison of the segmentation results obtained by 3 different models on the MoNuSeg testing set

(b) Comparison of the segmentation results obtained by 3 different models on the TNBC testing set

**Fig 4. Comparison of the segmentation results obtained by 3 different models on the testing set of the MoNuSeg and the TNBC.**

In our proposed MA-Net, the aggregation of multiple modules is not a simple additive relationship. Each module has its own strengths and weaknesses. The aggregation of these five modules compensates for the shortcomings of individual modules and highlights their respective advantages. As a result, it forms a new model that achieves stronger segmentation performance by leveraging the strengths of each module.

In order to more intuitively illustrate the advantages of MA-Net in nuclear segmentation, we randomly selected 3 images from the testing set of the MoNuSeg dataset and the TNBC dataset respectively, and displayed the segmentation results comparison between MA-Net and other methods in Fig 4. In these figures, the red box represents the locally enlarged image, and the orange box is the detail we focus on.

After being preprocessed, there are 120 images in the MoNuSeg dataset and 39 images in the TNBC dataset for model training. In supervised deep learning, the more training sets, the better results that model will produce to a certain degree. In Fig 4, it can be seen from the details of the display segmentation results figure that more training sets (the MoNuSeg training sets) can better enable the models to learn more feature information and produce better segmentation performance in the details.

While in the TNBC dataset, due to fewer training sets, most models could roughly predict the cell nuclei, it is not enough for the models to deal with the problems like multiple nuclear overlaps and unclear boundaries. The MA-Net we constructed still shows stable and accurate segmentation ability when there are few training sets, which proves that the model does have good robustness.

Compared with the detail in the orange boxes, we can see that in Fig 4A, Attention-UNet and CE-Net both have false recognition of smaller nuclei and a small number of intercellular adhesions exist. In Fig 4B, U-Net and UNet++ both have intercellular adhesions, while MA-Net shows a strong ability to deal with the problems of multiple nuclear overlaps, intercellular adhesions, unclear nuclear boundaries, and smaller nuclei misprediction.

**Table 2. Comparison of segmentation results of all the experimental methods on the MoNuSeg testing set and the TNBC testing set.**

| Model | Pre(↑) | | Acc(↑) | | Dice(↑) | | MIoU(↑) | |
|---|---|---|---|---|---|---|---|---|
| | MoNuSeg | TNBC | MoNuSeg | TNBC | MoNuSeg | TNBC | MoNuSeg | TNBC |
| U-Net[15] | 0.7013 | 0.7695 | 0.8864 | 0.9238 | 0.7393 | 0.7383 | 0.7235 | 0.7532 |
| SegNet[34] | 0.6621 | 0.4799 | 0.8699 | 0.8479 | 0.7308 | 0.5208 | 0.7185 | 0.5945 |
| Attention-UNet[20] | 0.7332 | **0.7966** | 0.9056 | 0.9226 | 0.7680 | 0.6902 | 0.7602 | 0.7320 |
| CE-Net[22] | 0.7185 | 0.7910 | 0.8976 | 0.9271 | 0.7647 | 0.7397 | 0.7439 | 0.7557 |
| BiO-Net[23] | 0.7728 | 0.7539 | 0.9142 | 0.9323 | 0.7769 | 0.7813 | 0.7682 | 0.7811 |
| UNet++[21] | 0.7614 | 0.7919 | 0.9136 | 0.9272 | 0.7801 | 0.7384 | 0.7726 | 0.7573 |
| DU-Net[35] | 0.7218 | 0.7628 | 0.9040 | 0.9022 | 0.7664 | 0.5813 | 0.7583 | 0.6618 |
| MA-Net(Ours) | **0.7803** | 0.7948 | **0.9204** | **0.9413** | **0.7936** | **0.8085** | **0.7826** | **0.8047** |

In this cell nuclei segmentation task, the method we proposed is compared to the vanilla U-Net and other state-of-the-art methods. Our model is improved according to the ideas of several other models and harmoniously integrates the advantages of these models. Therefore, it can better extract the feature map of nuclei in the encoder path, and in the decoder path, the image details that may be lost during down-sampling are better restored with the help of concatenating connection and our proposed up-sampling fusion module. It did achieve a higher evaluation score than the other seven models. MA-Net achieves 0.7948 in Precision score, 0.9413 in Accuracy score, 0.8085 in Dice score, and 0.8047 in MIoU score on the MoNu-Seg testing set and 0.7803 in Precision score, 0.9204 in Accuracy score, 0.7936 in Dice score, and 0.7826 in MIoU score on the TNBC testing set.

Table 2 shows the evaluation scores of eight cell nuclei segmentation models on the MoNu-Seg dataset and TNBC dataset respectively. As one can see, the method we proposed achieves a remarkable improvement and the best evaluation results on almost every evaluation indicator on the MoNuSeg testing set. Specifically, our method achieves 0.0639 higher than SegNet [34] in MIoU score and 0.0790 higher than the vanilla U-Net in Precision score. Also, it achieves 0.0289 higher than CE-Net and 0.0135 higher than UNet++, the best-known U-Net variant in Dice score. On the TNBC testing set, our method still achieves the greatest score on almost every evaluation indicator, although the score gap with other methods is not so significant as the experiment results on the MoNuseg testing dataset. Specifically, MA-Net achieves 0.0515 points higher than the vanilla U-Net in MIoU score and about 0.02 higher than UNet++, CE-Net, and Attention U-Net in Accuracy score. Also, it achieves over 0.2 higher than SegNet and DU-Net [35] in Dice score. Although MA-Net did not get the best result in Precision score, it still ranked in second place. The TNBC dataset consists of 50 histopathology images, it is a relatively small dataset compared with the MoNuSeg dataset. The score of the evaluation result on the MoNuSeg is lower than that on TNBC on average. We could say that the MoNu-Seg dataset is more challenging, while the TNBC dataset could test the robustness of the models according to the unstable evaluation result of SegNet and DU-Net. The segmentation task on the TNBC dataset is relatively easier for most model.

In addition to the above evaluation indicators, we also used ROC curves to evaluate all the models, which is shown in Fig 5 In the ROC coordinates, the closer the ROC curve to the upper left corner, the more accurate the corresponding model is. In another way, the accuracy of the ROC curve can be more clearly judged by the area under the ROC curve (AUC) score, which is also marked in the lower right corner of the figure. As one can see, the AUC scores of our MA-Net in both datasets are the highest among all the models, achieving 0.9667 and 0.9730 respectively.

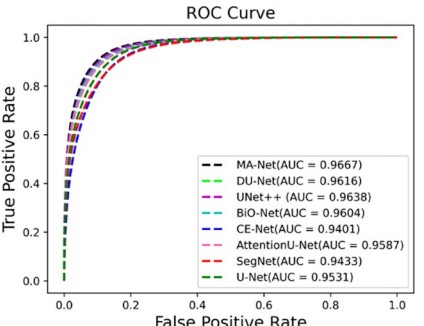 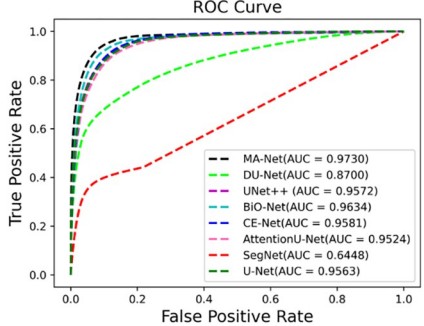

**(a) ROC curves on the testing set of the MoNuseg dataset** **(b) ROC curves on the testing set of the TNBC dataset**

**Fig 5. ROC curves in different models.** (a) ROC curves on the testing set of the MoNuseg dataset; (b) ROC curves on the testing set of the TNBC dataset.

We also compared MA-Net with advanced models based on self-attention mechanisms such as TransUnet [36]. Among them, TransUnet utilizes the Transformer for feature extraction in the encoder part. Due to the relatively small size of the dataset we used, making it difficult to train from scratch. thus, we initialized it with imagenet21k pre-trained parameters. UCTransNet [37] solely employs the Transformer module to process skip connections and does not use it for feature extraction. Hence, we directly trained UCTransNet without using pre-trained parameters, and the same applies to DCA-UNet [38] and MaxVit-Unet [39].

As shown in Table 3, except for MaxVit-Unet achieving slightly higher precision than MA-Net on the TNBC dataset, all other metrics were lower than those of MA-Net. The fundamental module employed in MaxVit-Unet is based on multi-axis attention [40], a sparse attention mechanism characterized by linear computational complexity and theoretically possessing a global receptive field. MA-Net is entirely implemented based on convolutional neural networks (CNNs) has relatively lower requirements for data volume during the training process. Therefore, it has an advantage when working with small-scale datasets.

Additionally, we tested our method on more datasets, including CryoNuSeg [41], BBBC039 [42] and kaggle 2018 data science bowl [43], all of which are nuclei segmentation datasets. We specifically compared Attention-UNet, CE-Net, and MaxVit-Unet with our MA-Net. These three models correspond to the common improvement strategies for UNet, namely enhancing the skip connection structure, improving the bottleneck block, and refining both the upsampling and downsampling paths, respectively.

The comparison results are shown in Table 4. Despite all three datasets being nuclear segmentation datasets, they differ in staining styles, cell densities, and image resolutions. Our model consistently achieved the best performance across all datasets, with the most significant

**Table 3. Comparison of MA-Net with attention-based state-of-the-art methods.**

| Model | Pre(↑) | | Acc(↑) | | Dice(↑) | | MIoU(↑) | |
|---|---|---|---|---|---|---|---|---|
| | MoNuseg | TNBC | MoNuseg | TNBC | MoNuseg | TNBC | MoNuseg | TNBC |
| TransUnet[36] | 0.779 | 0.799 | 0.906 | 0.932 | 0.788 | 0.781 | 0.768 | 0.781 |
| UCTransNet[37] | 0.752 | 0.791 | 0.903 | 0.924 | 0.787 | 0.732 | 0.766 | 0.755 |
| DCA-UNet[38] | 0.716 | 0.759 | 0.877 | 0.896 | 0.732 | 0.765 | 0.718 | 0.748 |
| MaxVit-Unet[39] | 0.756 | **0.836** | 0.906 | 0.940 | 0.791 | 0.766 | 0.772 | 0.781 |
| MA-Net(Ours) | **0.780** | 0.795 | **0.920** | **0.941** | **0.794** | **0.809** | **0.783** | **0.805** |

**Table 4. Comparison results on supplementary datasets.**

| Dataset | Model | Pre(↑) | Acc(↑) | Dice(↑) | MIoU(↑) |
|---|---|---|---|---|---|
| CryoNuseg | Attention-UNet | 0.748 | 0.892 | 0.754 | 0.734 |
| | CE-Net | 0.706 | 0.888 | 0.761 | 0.734 |
| | MaxVit-Unet | 0.781 | 0.888 | 0.737 | 0.722 |
| | Ours | **0.781** | **0.901** | **0.769** | **0.750** |
| BBBC039 | Attention-UNet | 0.969 | 0.987 | 0.971 | 0.963 |
| | CE-Net | 0.970 | 0.986 | 0.970 | 0.962 |
| | MaxVit-Unet | 0.976 | **0.988** | 0.972 | 0.965 |
| | Ours | **0.976** | 0.987 | **0.973** | **0.965** |
| Data Science Bowl-2018 | Attention-UNet | 0.854 | 0.958 | 0.833 | 0.849 |
| | CE-Net | 0.894 | 0.955 | 0.824 | 0.842 |
| | MaxVit-Unet | 0.894 | 0.962 | 0.850 | 0.867 |
| | Ours | **0.894** | **0.964** | **0.852** | **0.869** |

improvement observed on the CryoNuSeg dataset. This demonstrates that MA-Net possesses strong generalization capabilities and can effectively adapt to the task of nuclear segmentation.

Table 5 lists the training speed and the number of parameters for each model. Although MA-Net achieves the best segmentation performance among the models, it has a larger number of parameters and requires longer training times. Future work will focus on reducing the model parameters and increasing computational speed while maintaining segmentation performance. This can be achieved by further improving the network structure using techniques such as depthwise separable convolutions and group convolutions.

## Conclusions

Computer-aided automatic Cell Nuclei Segmentation is significant in medical image recognition and histopathological diagnosis assistance. In this paper, we have proposed a novel improved U-Net method, named MA-Net, used for cell nuclei segmentation. Compared with the vanilla U-Net structure, the proposed MA-Net aggregates the advantages of the multi-modules including, strengthened the contracting path by the Down-sampling Fusion Module, the expansive path by the Up-sampling Fusion Module, the skip-connection part by the

**Table 5. Comparison of the specific parameters of all the experimental methods.**

| Model | MoNuSeg | TNBC | Params (M) | model size (MB) |
|---|---|---|---|---|
| | Time for training (s/epoch) | time for training (s/epoch) | | |
| U-Net | 13.98 | 4.56 | 34 | 131 |
| SegNet | 10.55 | 3.05 | 29 | 112 |
| Attention-UNet | 12.19 | 5.25 | 34 | 133 |
| CE-Net | 4.49 | 2.37 | 40 | 154 |
| BiO-Net | 20.20 | 4.87 | 59 | 228 |
| UNet++ | 22.56 | 8.28 | 36 | 138 |
| DU-Net | 23.20 | 8.41 | 60 | 230 |
| TransUnet | 25.27 | 11.46 | 101 | 404 |
| UCTransNet | 24.00 | 8.91 | 64 | 257 |
| DCA-UNet | 20.00 | 6.57 | 42 | 170 |
| MaxVit-Unet | 18.37 | 4.48 | 24 | 97 |
| MA-Net(Ours) | 24.84 | 10.36 | 132 | 505 |

Attention Gate Module, the Context Extractor Module and the Residual Module. MA-Net can obtain more original information from inputs images, obtain more information in deep-level and retain spatial information and use the information for image segmentations more precisely. The results of our experiment on the MoNuSeg and TNBC datasets show that MA-Net is able to improve the cell nuclei segmentation effect, achieving 0.7826 and 0.8047 in MIoU score on the above datasets, which is 0.0591 and 0.0515 higher than vanilla U-Net. Also, it achieved the highest AUC score of 0.9667 and 0.9730 on the above typical datasets. It turns out that our method has reasonable segmentation performance and generalization ability. However, MA-Net also has its shortcoming that it spends more time in the training phase than the traditional one. MA-Net could be applied to a new semantic field as long as using the new training images and the corresponding labeled ground truth to train it. Our work demonstrates that, compared to improving a single part of the U-Net, simultaneously optimizing the structures of the encoder, decoder, skip connections, and bottleneck is more beneficial for feature extraction and fusion, thereby enhancing model performance. Future research will focus on model lightweighting, aiming to reduce the number of parameters while maintaining model performance.

## Author Contributions

**Conceptualization:** Qiumei Pu, Lina Zhao.

**Formal analysis:** Jinglong Tian, Donghao Wei.

**Funding acquisition:** Qiumei Pu, Lina Zhao.

**Methodology:** Qiumei Pu, Lina Zhao.

**Project administration:** Qiumei Pu, Lina Zhao.

**Software:** Jinglong Tian, Donghao Wei.

**Supervision:** Qiumei Pu, Lina Zhao.

**Writing – original draft:** Jinglong Tian, Donghao Wei.

**Writing – review & editing:** Qiumei Pu, Jinglong Tian, Donghao Wei, Qingming Shu, Minghui Sun, Lina Zhao.

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
