## [Decision Letter · Decision Letter 0]

8 Mar 2024

PONE-D-24-00318Multifunctional Aggregation Network of Cell Nuclei Segmentation Aiming Histopathological Diagnosis Assistance: A New MA-Net ConstructionPLOS ONE

Dear Dr. Zhao,

Thank you for submitting your manuscript to PLOS ONE. After careful consideration, we feel that it has merit but does not fully meet PLOS ONE’s publication criteria as it currently stands. Therefore, we invite you to submit a revised version of the manuscript that addresses the points raised during the review process.

We look forward to receiving your revised manuscript.

Kind regards,

Xiaowei Li

Academic Editor

PLOS ONE

Journal Requirements:

2. Thank you for stating the following financial disclosure: "National Key Research and Development Program of China (2021YFA1200904 and 2020YFA0710700), the National Natural Science Foundation of China (31971311, 12375326), the Innovation Program for IHEP (E35457U210)."

3. Thank you for stating the following in the Acknowledgments Section of your manuscript: "This work was supported by the National Key Research and Development Program of China (2021YFA1200904 and 2020YFA0710700), the National Natural Science Foundation of China (31971311, 12375326), the Innovation Program for IHEP (E35457U210)."

Please remove any funding-related text from the manuscript and let us know how you would like to update your Funding Statement. Currently, your Funding Statement reads as follows: "National Key Research and Development Program of China (2021YFA1200904 and 2020YFA0710700), the National Natural Science Foundation of China (31971311, 12375326), the Innovation Program for IHEP (E35457U210)."

Reviewers' comments:

Reviewer's Responses to Questions

**Comments to the Author**

1. Is the manuscript technically sound, and do the data support the conclusions?

Reviewer #1: Yes

Reviewer #2: Yes

2. Has the statistical analysis been performed appropriately and rigorously? 

Reviewer #1: Yes

Reviewer #2: Yes

3. Have the authors made all data underlying the findings in their manuscript fully available?

Reviewer #1: Yes

Reviewer #2: Yes

4. Is the manuscript presented in an intelligible fashion and written in standard English?

Reviewer #1: Yes

Reviewer #2: No

5. Review Comments to the Author

Reviewer #1: The paper introduces a method U-Net based model, MA-Net(Multifunctional Aggregation Network), to accurately segmenting nuclei and the model utilized the subsampling fusion module and upsampling fusion module on the U-Net encoder and decoder respectivel. Overall, the manuscript has a good structure and better validation of results via ablation studies. However, I suggest the manuscript to undergo major revision and resubmit for another quick review.

The following are some major corrections that can be incorporated to improve quality:

1) More experiments on several other datasets such as Fluorescence Microscopy Image Dataset and any other medical image segmentation datasets has to be performed to effectively understand the contribution of the proposed approach and validation of methodology. I suggest authors to make more evaluation on similar datasets.

2) Similarly, a detailed comparison on several recent methods (papers published on or after 2020) such as TransNuSeg, CellViT, has to be performed. The Authors can refer to https://paperswithcode.com/sota/medical-image-segmentation-on-monuseg for more details on the state-of-the-art methods.

3) The Abstract is too long and can be condensed highlighting the novelty and best results on datasets.

4) The recommend authors to highlight the novelty in methodology. The manuscript is very similar to CE-Net (Context Encoder Network for 2D Medical Image Segmentation) thereby might be a further extension to cell nuclei segmentation application.

5) I suggest authors to add future scope for further improvements and research in this direction and add and explain a figure for some of the failure cases of using this methodology.

Minor corrections:

1) Extra full stop on the first line in Abstract. I recommend Authors to carefully read and review the manuscript for any typographical errors.

2) Equations can be enhanced in the methodology. I suggest authors to use LaTeX while formulation equations.

3) Hyperlinks has to be added to citations and references for helping readers to quickly refere the cited studies.

Reviewer #2: Dear Editor, I want to thank you very much for the invitation to review a manuscript entitled "Multifunctional Aggregation Network of Cell Nuclei Segmentation Aiming Histopathological Diagnosis Assistance: A New MA-Net Construction" for your journal. In fact, the authors raised an assessment about a U-Net based deep learning model, called MA-Net(Multifunctional Aggregation Network), to accurately segmenting nuclei from H&E stained images. The improved model utilized the subsampling fusion module and upsampling fusion module on the U-Net encoder and decoder respectively, to better restore the detailed information lost during the image encoding process and avoid blurry boundaries in segmentation result. however, I have some minor points:

comment 1: What about the accrual result of this technique with different types of cancers? is it accurate?

comment 2: "Experimental results substantiate that our proposed MA-Net outperforms other improved models in terms of cell nuclei segmentation tasks" I think this sentence is confusing, how can you support this information?

Comment 3: Some editing for English language is required throughout the manuscript due to too many mistakes.

6. PLOS authors have the option to publish the peer review history of their article (what does this mean?). If published, this will include your full peer review and any attached files.

Reviewer #1: No

Reviewer #2: **Yes: **Mohamed Hadi Mohamed Abdelhamid

---

## [Author Response · Author response to Decision Letter 0]

21 Jun 2024

Response to reviewers

Major Corrections

1.Comment: More experiments on several other datasets such as Fluorescence Microscopy Image Dataset and any other medical image segmentation datasets has to be performed to effectively understand the contribution of the proposed approach and validation of methodology. I suggest authors to make more evaluation on similar datasets.

Reply: 

Thank you for your suggestions. We have conducted additional experiments using more datasets to further demonstrate the effectiveness of our proposed method. The results and analysis of these experiments have been added to Section 4.3 "Results and Discussion" in the main text.

2.Comment: Similarly, a detailed comparison on several recent methods (papers published on or after 2020) such as TransNuSeg, CellViT, has to be performed. The Authors can refer to https://paperswithcode.com/sota/medical-image-segmentation-on-monuseg for more details on the state-of-the-art methods.

Reply:

Thank you for your suggestion. To further validate the effectiveness of our proposed method, we conducted additional experiments using state-of-the-art Transformer-based approaches post-2023 on the MoNuSeg and TNBC datasets. The results and analysis of these experiments are presented in Section 4.3, Results and Discussion.

3.Comment: The Abstract is too long and can be condensed highlighting the novelty and best results on datasets.

Reply:

Thank you for reviewing our paper and providing your suggestions. We understand that the abstract is an essential part for readers to quickly grasp the research content and outcomes. We have made the necessary simplifications and optimizations to the abstract in the revised manuscript.

4.Comment: The recommend authors to highlight the novelty in methodology. The manuscript is very similar to CE-Net (Context Encoder Network for 2D Medical Image Segmentation) thereby might be a further extension to cell nuclei segmentation application.

Reply:

Thank you for reviewing our paper and for your valuable comments. While both our work and CE-Net focus on semantic image segmentation, and thus inevitably share some similarities, it is important to emphasize that our work is not an extension or continuation of CE-Net. Our design approach has several key differences:

Firstly, CE-Net's improvements mainly focus on the Bottleneck part of the U-Net network, whereas our work involves improvements to the encoder, decoder, skip connections, and Bottleneck of the U-Net. We conducted comparative experiments using CE-Net as a baseline model, and the results demonstrate the effectiveness of our proposed feature fusion strategy and spatial attention mechanism. Secondly, our work differs from CE-Net in terms of training setup. Considering the characteristics of cell nucleus medical imaging, we adopted a hybrid loss function different from CE-Net and employed different data augmentation methods. Lastly, in terms of application, CE-Net is applied to retinal vessel images, while our work primarily targets cell nucleus segmentation.

Regarding your concerns about the novelty of our method, we have revised the abstract and relevant sections of the main text to further emphasize the originality and novelty of our work.

5.Comment: I suggest authors to add future scope for further improvements and research in this direction and add and explain a figure for some of the failure cases of using this methodology.

Reply:

Thank you for your suggestions. We also recognize the importance of analyzing the current limitations and considering improvements for future work. We have added relevant content at the end of Section 4.3 "Results and Discussion" and in Section 5 "Conclusions".

Minor Corrections

1.Comment: Extra full stop on the first line in Abstract. I recommend Authors to carefully read and review the manuscript for any typographical errors.

Reply:

Thank you for pointing out the issues. We have carefully reviewed the manuscript of the article and made corrections to the formatting and textual errors.

2.Comment: Equations can be enhanced in the methodology. I suggest authors to use LaTeX while formulation equations.

Reply:

Thank you for your suggestions. We have carefully reviewed and verified the formulas in the text, and we have used MathType to rewrite them, correcting any formatting and layout errors.

3.Comment: Hyperlinks has to be added to citations and references for helping readers to quickly refere the cited studies.

Reply:

Thank you for your suggestion. We have added hyperlinks to all the references cited in the article. Readers can now click on the citation numbers in the text to navigate to the corresponding references.

Special thanks to you for your good comments.

---

## [Decision Letter · Decision Letter 1]

23 Jul 2024

Multifunctional Aggregation Network of Cell Nuclei Segmentation Aiming Histopathological Diagnosis Assistance: A New MA-Net Construction

PONE-D-24-00318R1

Dear Dr. Zhao,

We’re pleased to inform you that your manuscript has been judged scientifically suitable for publication and will be formally accepted for publication once it meets all outstanding technical requirements.

Kind regards,

Xiaowei Li

Academic Editor

PLOS ONE

Additional Editor Comments (optional):

Reviewers' comments:

Reviewer's Responses to Questions

**Comments to the Author**

1. If the authors have adequately addressed your comments raised in a previous round of review and you feel that this manuscript is now acceptable for publication, you may indicate that here to bypass the “Comments to the Author” section, enter your conflict of interest statement in the “Confidential to Editor” section, and submit your "Accept" recommendation.

Reviewer #1: All comments have been addressed

Reviewer #2: All comments have been addressed

2. Is the manuscript technically sound, and do the data support the conclusions?

Reviewer #1: Yes

Reviewer #2: Yes

3. Has the statistical analysis been performed appropriately and rigorously? 

Reviewer #1: Yes

Reviewer #2: Yes

4. Have the authors made all data underlying the findings in their manuscript fully available?

Reviewer #1: Yes

Reviewer #2: No

5. Is the manuscript presented in an intelligible fashion and written in standard English?

Reviewer #1: Yes

Reviewer #2: Yes

6. Review Comments to the Author

Reviewer #1: I thank the Authors for thoroughly and carefully revising the manuscript and it is now much better, and interesting for the readers.

Reviewer #2: Dear Editor and Authors,

Thank you for your work on this manuscript. We appreciate your efforts in addressing all the comments.

7. PLOS authors have the option to publish the peer review history of their article (what does this mean?). If published, this will include your full peer review and any attached files.

Reviewer #1: No

Reviewer #2: **Yes: **Mohamed Hadi Mohamed Abdelhamid

---

## [Editor Report · Acceptance letter]

30 Jul 2024

PONE-D-24-00318R1 

PLOS ONE

Dear Dr. Zhao, 

I'm pleased to inform you that your manuscript has been deemed suitable for publication in PLOS ONE. Congratulations! Your manuscript is now being handed over to our production team.

Kind regards, 

on behalf of

Dr. Xiaowei Li 

Academic Editor

PLOS ONE